# Nanoscale CAR Organization at the Immune Synapse Correlates with CAR-T Effector Functions

**DOI:** 10.3390/cells12182261

**Published:** 2023-09-12

**Authors:** Julia Sajman, Oren Yakovian, Naamit Unger Deshet, Shaked Almog, Galit Horn, Tova Waks, Anat Globerson Levin, Eilon Sherman

**Affiliations:** 1Racah Institute of Physics, The Hebrew University, Jerusalem 91904, Israel; 2Jerusalem College of Technology, Jerusalem 91160, Israel; 3Immunology and Advanced CAR-T Cell Therapy Laboratory, Research & Development Department, Tel-Aviv Sourasky Medical Center, Tel Aviv 6423906, Israel; 4Dotan Center for Advanced Therapies, Tel-Aviv Sourasky Medical Center and Tel Aviv University, Tel Aviv 6423906, Israel

**Keywords:** CAR-T, CD138, super resolution microscopy, immune synapse, CD45, nanoclusters, cancer, adoptive cell immunotherapy

## Abstract

T cells expressing chimeric antigen receptors (CARs) are at the forefront of clinical treatment of cancers. Still, the nanoscale organization of CARs at the interface of CAR-Ts with target cells, which is essential for TCR-mediated T cell activation, remains poorly understood. Here, we studied the nanoscale organization of CARs targeting CD138 proteoglycans in such fixed and live interfaces, generated optimally for single-molecule localization microscopy. CARs showed significant self-association in nanoclusters that was enhanced in interfaces with on-target cells (SKOV-3, CAG, FaDu) relative to negative cells (OVCAR-3). CARs also segregated more efficiently from the abundant membrane phosphatase CD45 in CAR-T cells forming such interfaces. CAR clustering and segregation from CD45 correlated with the effector functions of Ca^++^ influx and target cell killing. Our results shed new light on the nanoscale organization of CARs on the surfaces of CAR-Ts engaging on- and off-target cells, and its potential significance for CAR-Ts’ efficacy and safety.

## 1. Introduction

T cells search for cognate antigens for mounting an appropriate immune response, while robustly rejecting self-antigens. The cognate antigens are processed and presented by specialized cells, known as antigen presenting cells (APCs). Upon recognition, the T cells form a tight interface with the APCs, known as the immune synapse (IS) [1,2]. The nanoscale organization of receptors and surface molecules at the IS has been shown to affect T cell activation [3,4,5]. An outstanding example involves the induced self-clustering of TCRs upon engagement of cognate antigens [6,7,8]. In another example, the tight interface between the cells at the IS has been shown to mechanically exclude bulky surface glycoproteins, such as CD45, CD48 and CD148, while TCRs can bind peptide-MHC ligands in these tight contacts. Since such glycoproteins serve as phosphatases, they tend to dephosphorylate tyrosines on specific immunoreceptor tyrosine-based activation motifs (ITAMs) on the TCR, and thus may quench the TCR signal. It has been suggested and shown that the physical segregation of such glycoproteins by the tight interfaces at the IS may allow a more efficient TCR signaling. This mechanism, called the ‘kinetic segregation’ (KS) model [9], was also shown to occur already at early T cell contacts that precede the mature IS with APCs [10,11].

T cells expressing chimeric antigen receptors (CAR-Ts) are an emerging tool for cancer treatment, and hold the promise for targeting a wide variety of cancers and other diseases (e.g., AIDS) [12]. The TCR is constructed of multiple chains (α, β, γ, δ, ε, ζ), whereas CARs are single-chained receptors by design. They incorporate a sc-Fv based on the variable region of a specific antibody, for specific recognition of antigens, independently of the major histocompatibility complex (MHC). Downstream to the sc-Fv, and in order to separate it from the cell surface, is a hinge, followed by a trans-membrane domain, intracellular domains that contain one or multiple signaling domains and an immunoreceptor tyrosine-based activation motif (ITAM) for signal transduction.

While the structure of CARs has been carefully designed, the nanoscale organization of CARs at the interface of CAR-Ts with target cells remains poorly understood. For instance, the extent of self-clustering and the ability of CARs to segregate bulky glycoproteins (as achieved by TCRs) remain unclear. This is especially true since CAR-Ts may not require well-structured synapses for engagement and killing of target cancer cells [12,13].

Here, we studied the nanoscale organization of CARs targeting CD138 (Syndecan-1) and their relation to surface glycoproteins. We employed single- and two-color single-molecule localization microscopy (SMLM) to resolve individual molecules on the surface of the CAR-T cells, as they engaged either on-target or off-target cancer cells. Our imaging further included a technique, recently developed in our lab, for aberration-free SMLM imaging of the interface between the cells (fixed or alive) with resolution down to ~20 nm [14]. We discovered that CARs showed significant self-clustering that was enhanced in interfaces with on-target cells (SKOV-3, CAG, FaDu) relative to negative cells (OVCAR-3). CARs also segregated more efficiently from CD45 in CAR-T cells forming such interfaces. CAR clustering and segregation from CD45 correlated with the effector functions of Ca^++^ influx and target cell killing.

Altogether, our results shed a new light on the nanoscale organization of CARs on the surfaces of CAR-T cells. This organization could serve as an early and sensitive marker for potential CAR-T efficacy and safety, informing the optimization of CAR designs against various cancers.

## 2. Materials and Methods

### 2.1. Primary Cells and Cell Lines

Untreated (UT) lymphocyte primary cells and CD138-CAR-T primary cells;FaDu cell lines—Human pharynx, squamous epithelial carcinoma.SKOV-3 cell lines—Human ovary epithelial adenocarcinoma, were a kind gift from the Dan Peer lab at Tel Aviv University;CAG—Human multiple myeloma cell line;OVCAR-3—Human ovarian adenocarcinoma.

Unless specified otherwise, all cells were provided by the Immunology and advanced CAR-T cell therapy Laboratory, Research & Development Department, Tel-Aviv Sourasky Medical Center, Tel Aviv, Israel.

### 2.2. Cell Growth and Treatment

Cell lines (FaDu, SKOV-3, CAG and OVCAR-3) grow in RPMI + 10%FBS + 1% Pen/Strep. Primary UT and CAR-T cells grown in RPMI + 10%FBS + 1%Pen/Strep + 100 units IL-2. CAR-T cells were prepared according to a previous study [15].

### 2.3. Membrane Staining

The plasma membrane was tagged by incubation of the cells in staining solution containing 10 μM DiD or DiO (Vybrant^®^ DiD Cell-Labeling Solution, Invitrogen, Waltham, MA, USA, V22887), in PBS for 5–10 min. After staining, cells were washed according to the manufacturer’s protocol.

### 2.4. Confocal Imaging

The imaging was performed on an FV-1200 confocal microscope (Olympus, Tokyo, Japan).

### 2.5. Cell Immuno-Staining

Antibodies were used following the manufacturers’ protocols. Briefly, 0.5 µg of primary fluorescent conjugated antibody was added to 500 × 10^3^ cells suspended in FACS buffer (90% PBS 10% FBS 0.02% Na-Azide) for 30–60 min on ice. Cells were washed and suspended in imaging buffer (RPMI without phenol red, 10% FBS, 25 mM HEPES) for live cell imaging.

The antibodies used in our experiments include:Mouse Monoclonal Strep Tag II Antibody-FITC conjugated (clone 5A9F9) LS-C203631;Mouse monoclonal IgG1 α-Human CD45-Alexa647 conjugated (BioLegend, San Diego, CA, USA, 304056);Mouse monoclonal IgG1 α-Human CD138—APC conjugated (BioLegend, 352308).

### 2.6. Microscope Slide Preparation for Cell-on-Cell

For visualizing conjugated cells in a horizontal orientation, we attached the different types of cells to different glass surfaces that were placed one on top of the other. Two types of glass surfaces were prepared: the 8-well glass chambers and small glasses that fit the opening of the chamber well. Coverslips were incubated with 10 μg mL^−1^ non-stimulatory antibodies (Mouse anti human CD45 (BD Pharmingen, San Diego, CA, USA, PMG555480); Mouse monoclonal IgG2a αCD11a (LFA1α) (BD Pharmingen, 555378)) overnight at 4 °C or 2 h at 37 °C. Finally, coverslips were washed with phosphate-buffered saline (PBS).

Adherent tumor cells were seeded the day prior to the experiment in a chamber well. The lymphocytes cells were dropped on an opposite small glass coated with αCD45/αCD11 antibody for 10 min in 37 °C for attachment. We then flipped the small glass on the adherent cells in the chamber well. In order to keep an appropriate space between two glasses we used 20 µm standard silica beads (47148-10, Corpuscular, Cold Spring, NY, USA). In fixed cell imaging, cells after staining and dropping, were fixed by 2.4% Paraformaldehyde (PFA) for 30 min in 37 °C and washed with PBS.

### 2.7. Single-Molecule Localization Microscopy

Two-color dSTORM imaging was conducted both for fixed and live cells. Cells were suspended in a STORM imaging buffer [16,17]. Imaging cells on coverslips was performed using a Nikon microscope with a CFI Apo TIRF X100 oil objective (NA 1.49, WD 0.12 mm).

Imaging of cells on cells was performed above the coverslip without using the ‘perfect focus’ feature of the microscope. Fluorophores were imaged in a following frame using laser excitation at either 488 nm, or 647 nm (~50% of 90 mW maximum for 488 nm or 200 mW for 647 nm). Laser illumination at all wavelengths covered a circular area with a diameter of 80 µm at the sample. dSTORM acquisition sequence typically took ~2.5 min at 13.4 fps of an EMCCD Ixon+ camera. The pixel size was equivalent to 160 × 160 nm^2^ at the sample. Excitation and imaging were performed through a quad dichroic (C-NSTROM QUAD 405/488/561/647/FILTER; Nikon, Tokyo, Japan).

### 2.8. dSTORM Reconstruction

Data acquired by SMLM were analyzed by a previously described algorithm (ThunderSTORM) [18] for the identification of individual peaks and grouping them into functions that reflect the positions of single molecules [19]. Next, peaks were grouped and assigned to individual molecules for rendering. Peak grouping used a distance threshold and a temporal gap to account for possible molecular blinking [19]. For fixed cells experiments, a temporal gap of ~50 msec and a distance threshold of 20 nm were applied for each fluorophore separately in order to minimize possible over-counting of molecules. Notably, each antibody may carry more than a single fluorophore, which cannot be readily accounted for per localization. Thus, data should be best regarded as comparative between the positive and negative cancer cells. Drift compensation and channel registration were performed using dedicated algorithms in the ThunderSTORM software (v1.3). For live cells experiments no drift compensation was applied, and a distance threshold of 20 nm was taken (regarding time gaps, each image in a live experiment accumulates 2–2.5 s of acquisition time as will be described next). Calibration was conducted using 100 nm Tetraspeck fluorescent beads (Invitrogen). Individual molecules are presented in dSTORM images with intensities that correspond to the probability density values of their fitted Gaussian with respect to the maximal probability density values detected in the field.

To generate a frame in a live cell imaging sequence, 50 frames of an SMLM movie with a frame rate of 50 fps were acquired, with alternating acquisition of the green and red channels. Thus, each image represents 1 s of SMLM acquisition time. The images were assigned the frame time of the first participating frame from the SMLM movie. These accumulated frames were further used to generate movies of the cell conjugates.

### 2.9. Three-Dimensional dSTORM

Three-dimensional imaging was performed using a cylindrical lens. Data were analyzed after 3D calibration of the system, according to the ThunderSTORM plugin protocol in ImageJ and using this tool [19].

### 2.10. DBSCAN Cluster Size Analysis

In order to define self-clusters of CD45 and CAR, we analyzed the proteins’ localization through the DBSCAN algorithm [20]. This algorithm defines clusters using point pattern data with two input parameters: a minimum number of neighbors required to identify a core point (minpts) and a neighborhood search radius (epsilon). We determined minpts = 2 and epsilon = 45 nm, as recommended by Nieves et al. [21]. In this way, the code defined and separated CD45 and CAR proteins into groups of monomers, dimers and clusters of 3 m and more. Finally, we reported the cumulative distribution of cluster sizes.

### 2.11. Calcium Assay

For calcium-influx experiments, CD138-CAR-T live cells were loaded with Fluo-4AM (Molecular Probes, F10489) at 5 μM for 60 min in the presence of 2.5 mM probenecid. Stained CAR-T cells were engaged to different adherent cells. Imaging was performed in Ca^++^-rich imaging buffer. Imaging at a large scale was performed by acquiring grids of about 100 fields of view, with single-cell resolution. Imaging was performed using a TiE Nikon microscope using epifluorescent illumination. We quantified Fluo-4 responses by determining the average intensity of a region within each cell (minus the background) as a function of time using the ImageJ program (NIH).

### 2.12. FACS Analysis for CD138 Expression Level

Cell cultures, CAG, SKOV-3, FaDu and OVCAR-3, were detached from the growth plate and immunostained in suspension with α-Human CD138 antibody conjugated to APC. Expression of CD138 ligand was measured on the live cells by flow cytometry (CellStream Analyzer, Luminex, Austin, TX, USA). The florescence of the stained cells was compared to the matching unstained cells for each cell line.

### 2.13. Killing Analysis

To assess the killing efficiency of the CD138-CAR-T cells, they were incubated with different target cells. The target cells, CAG, SKOV-3, FaDu and OVCAR-3, were seeded a day prior to the experiment in 24-well plates (100,000 cell/well). Next day, CD138-CAR expressing T cells were added (10^6^ cell/well). After 24 h and 48 h, the cells were imaged using bright-field microscopy with ×10 magnification We noticed pronounced cell aggregation for on-target cells after 24 and 48 h post treatment, which we associated with cell death.

The bright-field images were analyzed using the Trainable Weka Segmentation (v3.3.1) algorithm in ImageJ. Specifically, features in the imaged fields were classified into 3 classes: empty areas, areas with intact cells and areas with aggregated cells. The classifier was first trained manually on easily identified features, and then applied at once for all the imaged fields in the experiment (N > 32 for each cell type and each condition). We note that the shown fraction is an underestimate of the actual fraction of the cells killed, since the aggregates extend in 3D, while intact cells adhere to the coverslip in our assay.

### 2.14. MTT Assay

The target cells, CAG, SKOV-3, FaDu and OVCAR-3, were seeded a day prior experiment in 96 flat-bottom well plates (10,000 cell/well). Next day, CD138-CAR expressing T cells were added (10^5^ cell/well). After 24 h and 48 h MTT assay was performed according to manufacturer instructions.

## 3. Results

### 3.1. High-Resolution Imaging of the Interface between CAR-T and Target Cells

We used our recently developed assay for super-resolution imaging of cell–cell interfaces as they adhere to opposing coverslips [14]. Briefly, T (or CAR-T) cells and target cells are adhered to two opposing surfaces that are brought into contact, while spherical beads serve as physical spacers between these surfaces. Importantly, there was no interference with intact cell activation [14]. Here, we followed the interaction of CD138-CAR-T cells (CD138-CAR) as they engaged target cells that exclusively express CD138. The CD138-CAR configuration is as follows: an anti-CD138 scFv, a STREP-tag, co-stimulation using a CD28 hinge and transmembrane domain, and FcγR as an ITAM (Figure 1A) [15].

CD138 (called also Syndecan-1) is an integral membrane protein which participates in cell proliferation, cell migration and cell–matrix interactions via its receptor for extracellular matrix proteins. Syndecan-1 binds multiple growth factors and chemokines [22], and mediates cell binding, cell signaling and cytoskeletal organization. Importantly, altered CD138 expression has been detected in several different tumor types [23]. Specifically, CD138 may serve as a surface marker expressed on both normal and malignant plasma cells, and thus has been considered for treatment of relapsed/refractory multiple myeloma [24].

In this study we used different malignant cell lines expressing various levels of CD138 (FACS analysis, Figure 1B). Human ovarian adenocarcinoma, OVCAR-3, shows no expression of CD138, and served here as a negative control, with no stimulation of CAR-T cells expected upon engagement. Human multiple myeloma cell line, CAG, expresses a high level of CD138 and was described previously as a target cell line for CD138-CAR [15]. Human ovarian adenocarcinoma, SKOV-3, and Human Pharyngeal Squamous Cell Carcinoma, FaDu, also show high expression of CD138. Since SKOV-3 cells are derived from ovarian cancer, they match the negative OVCAR-3 cells. In addition, SKOV-3 and FaDu serve as examples of malignant cells derived from solid tumors.

In order to image CAR-T cells’ interaction with target cancer cells, the CAR-T cells were adhered on the top coverslip while the cancer cells were adhered to the bottom coverslip (Figure 1C). The coverslips were brought into contact, and the cells were fixed and stained after ~10 min. The CAR-T cells and the target SKOV-3 cells (stained, respectively, with the fluorescent membrane labels DiO, in green, and DiD, in red), as well as their interface, could be clearly identified using confocal microscopy (Figure 1D–F). Cell–cell engagement was visualized by Z-stack images as well as a side view of the interaction. Figure 1F represents a 2D plane (in this example cell), in which we chose to visualize the immune synapse organization. We could also visualize the interface between these cells using two-color dSTORM imaging (Appendix A). Interestingly, such imaging emphasizes features of irregularity at the interface between the cells that could not be distinguished using diffraction-limited microscopy.

### 3.2. Imaging and Analyses of Nanoscale Molecular Organization at the Interface

Specifically, we were interested in resolving the nanoscale organization of CARs and their relation to CD45 glycoproteins at the surface of the engaged CAR-T cells. Unfortunately, confocal microscopy is limited by diffraction and cannot resolve the intricate nanoscale organization of these molecules. Thus, we used the method of two-color direct stochastic optical reconstruction microscopy (dSTORM [16]; a type of SMLM) to study the organization of CARs and CD45 on the surface of the CAR-T cells (compare fluorescence images in Figure 2 and in Appendix A; note that not all T cells in our assay expressed the CARs). For dSTORM visualization, individual CARs carried a STREP tag, and were labelled using αSTREP-FITC antibody. CD45 surface molecules were labelled using a primary antibody carrying Alexa647.

Figure 2A shows a single CAR-T cell (two-color labelled) engaging an unlabeled FaDu target cell spread on the bottom coverslip. Bright-field imaging (overlaid) allowed us to determine the engagement between CAR-T and the adherent target cell. Two-color dSTORM imaging could resolve single molecules at the interface between the cells (Figure 2B). For such imaging we first optimized the merging parameters of each fluorophore (see Section 2 and Appendix A) and calculated histograms of key localization parameters (Appendix A). Figure 2C,F provides zoom on the fluorescent image of CD138-CAR and CD45.

Our imaging showed a highly non-uniform distribution of both CD138-CARs and CD45 on the surface of the CAR-T cells (Figure 2B, green and red, respectively). CARs appeared in pronounced clusters, while CD45 was enriched along elongated lines (likely lamellae ridges). CAR clusters either co-localized or segregated from CD45 (Figure 2C and Figure 2F, respectively).

We then analyzed the (self-) clustering extent of each of these molecules (Figure 2D,G) using pair correlation functions (PCFs). These functions indicated significant clustering (colored curves) of both CD138-CARs and CD45, relative to a model of complete spatial randomness (CSR, yielding a PCF value of g(r) = 1; flat black lines).

To study the relative organization of CARs and CD45, we used bivariate PCFs [11]. It is useful to normalize these curves [25]. Specifically, a measure of extent of mixing (EOM; Figure 2E,H) normalizes the bivariate PCF in such a way that a value of 0 corresponds to no interaction, while a value of 1 corresponds to a model of random labelling (see Section 2; Appendix A). This latter model indicates close association of the two molecules under study [4,25]. Our results could capture the relatively high (co-localized) or low (segregated) interaction of some of the CARs and CD45 on the surface of the engaged CAR-T cells (Figure 2E,H, respectively).

Taken together, our results show that CARs form a non-trivial organization relative to CD45 on the surface of CAR-T cells as they engage target cancer cells.

We next wanted to test the organization of CARs and CD45 on the surface of CAR-T cells without the presence of a target cell. For that, we used similar two-color dSTORM and bright-field imaging of the lower surface of CD138-CAR-T cells, without target cells on the bottom coverslip (Figure 2I,J). We found that self-clustering of both CARs and CD45 was significantly enhanced in the non-engaged cells (Figure 2K), and the co-localization of these molecules was also enhanced (Figure 2L). We attribute these organization patterns to the highly lamellar topography of the non-engaged CAR-T cells, and the cross-sectioning of our imaging plane of this topography.

### 3.3. CARs Cluster and Segregate to a Higher Extent from CD45 in Interfaces with On-Target Cells

We next studied the nanoscale organization of CARs and CD45 on interfaces between the CAR-T cells and various on- and off-target cells. For that, we conducted similar SMLM imaging of the interfaces between CAR-T cells and either OVCAR-3 (negative for CD138) or SKOV-3, CAG and FaDu (positive for CD138). As before, the cells were each attached to a slide and the slides were placed one on top of the other such that the cells were brought into contact and then fixed prior to imaging.

CARs demonstrated a higher extent of self-clustering on the on-target cells (in Figure 3, compare images in panel A for the negative OVCAR-3 cells with panels B–D for the positive cells). PCF statistics could capture the enhanced clustering of CARs (green curves) on the on-target cells relative to OVCAR-3 (compare panels E with F–H). CD45 (red curves) seemed self-clustered, yet without notable differences between the negative and positive CD138-expressing cells.

We further used a clustering algorithm [4], to characterize the self-clusters of CARs at the surface of the CAR-T cells as they engage the off- and on-target cells (Appendix A). We found that the cumulative histogram for all conditions was dominated by very small clusters, as small as dimers (~80–90%), trimers (5–7%), etc. Only a small fraction of the CAR clusters had dozens of molecules. We observed that CAR nanoclusters in ligand-expressing cancer cells show a larger fraction of the bigger clusters (from a few units, up to 1000 s of units) relative to the negative cancer cells (compare results for CAG, SKOV-3 and FaDu relative to OVCAR-3 (gray curves)).

Also, CARs segregated to a higher extent from CD45 upon CAR-T engagement of on-target cells, as indicated by the EOM statistics for each of the conditions (Figure 3, compare panel I with panels J–L).

T cells have in general an irregular surface with various protrusions. Previous studies have shown that surface molecules can organize differentially along such protrusions, known as microvilli [26,27]. Thus, we were interested in resolving the 3D component of the organization of CARs and CD45, including their clustering and segregation. For that, we employed 3D dSTORM of the CAR-T cells on the target and non-target cells (Appendix A; see Section 2). We found that CARs resided on average at higher planes relative to CD45 at the interface between the imaged cells (Appendix A). Strikingly, this height separation was highly significant (*p* = 5 × 10^−7^) in interfaces with the negative OVCAR-3 cells, while either not significant or of much lower significance in the positive cells (Appendix A). We attribute this observation to the formation of a tighter and more regular interface of the CAR-T cells with the positive cells.

Imaging of fixed cells could involve artefactual clustering of molecules that might affect the physiological patterns that occur at the interface between CAR-T cells and engage on- and off-target cells. Thus, we employed our imaging technique to study live cell conjugates of CAR-T cells and the on- and off-target cells (Figure 4). We note that the molecular patterns seemed slightly smoother in these images (compare Figure 4A–D to Figure 3A–D; esp. in panels B,C of both figures). Likewise, the PCF statistics still showed significant self-clustering of CARs and CD45 under all conditions, yet to a lower extent than for the fixed cells. Importantly, as found for fixed cells, CARs self-clustered to a higher extent on the on-target cells relative to the off-target cells (OVCAR-3) (compare Figure 4, panel E with panels F,G and H). Differences in segregation of CARs from CD45 were also evident (compare Figure 4, panel I with panels J–L).

The results of CAR self-clustering for fixed and live cells are summarized in Figure 5A. CAR clustering was significantly higher at the interface between CAR-T cells and on-target cells. The EOM results are summarized in Figure 5B,C for the fixed and live cells, respectively. The results are shown for EOM values at 20 nm (indicating close molecular overlap) and at 200 nm (indicating overlap of larger features, such as microvilli and lamellae ridges). CARs segregated significantly from CD45 upon engagement with the on-target SKOV-3, CAG (in fixed cells) and FaDu cells; all relative to the off-target cells OVCAR-3.

### 3.4. CAR–Glycoprotein Segregation Correlates with Key Effector Functions

Next, to study whether the nanoscale organization of CARs and CD45 correlates with the CAR-T cells’ activity, we measured the Ca^++^ influx of these cells upon engagement of the on-target or the off-target cells. The CAR-T cells demonstrated robust Ca^++^ influx upon encounter with the target SKOV-3, CAG and FaDu cells (see frames from movie taken on CAG cells; Figure 6A upper panel), but not with the negative OVCAR-3 cells (Figure 6A, lower panel).

We further visualized large sets of on- and off-target cells after engagement with the CAR-T cells for more extended durations of 24 and 48 h. Strikingly, on-target FaDu, SKOV-3 and CAG cells showed aggregation and floating at 24 h, indicating efficient killing by the CAR-T cells. In contrast, the off-target OVCAR-3 cells were unharmed by the CAR-T cells, even after 48 h. We estimated the number of killed cells by measuring the fraction of cells in the apparent aggregates (Figure 6C and Appendix A). Indeed, we found a significant fraction of aggregated (i.e., killed) cells for the on-target cells after 24 h. This fraction significantly decreased at 48 h for CAG and FaDu (but not for SKOV-3), likely due to cell proliferation after that prolonged time. The enhanced CAR-T killing of the on-target cells vs. the negative, off-target cells was further verified using a stain for cell viability (MTT; see Section 2) at the same time points (Figure 6D). In this assay, the cell killing was not reduced after 48 h, likely due to the presence of a large fraction of the CAR-T cells relative to the target cells.

## 4. Discussion

The nanoscale organization of signaling molecules on the surface of T cells has been shown to determine their activation downstream of the TCR [28]. In contrast, much less is known about such organization of the interface between CAR-Ts and target cancer cells. Here, we resolved the nanoscale organization of CARs targeting cancer-specific CD138 glycans in intact (fixed and live) cell conjugates, at the single-molecule level. Specifically, we studied the self-clustering of the CARs and their organization relative to the glycoprotein phosphatase CD45. CARs showed significant self-clustering at the interface with both on- and off-target cells yet had a higher extent of clustering upon engagement with the on-target cells. Also, CARs efficiently segregated from CD45 at the interface between CAR-T cells and on-target cells (SKOV-3, FaDu, CAG) relative to the off-target cells, lacking expression of CD138 (OVCAR-3).

To conclude our results, we studied the potential correlation between the relative ligand expression of the off- and on-target cells, the nanoscale parameters describing CAR clustering and segregation from CD45 and the effector functions of Ca^++^ influx and target cell killing. These parameters were each normalized by the maximal value observed for any of the cell lines. We found high correlation between all these parameters (Figure 7), with the mutual Pearson correlations detailed in Table 1 (note that correlation of the killing parameter with the rest of the parameters was lowered due to the delayed killing of SKOV-3 cells by the CAR-T cells). These correlations indicate that relative ligand expression on the target cells, CAR clustering and CAR-CD45 segregation correlate with the key effector functions of Ca^++^ influx and selective killing of on-target cells.

Some CARs have been shown to form self-clusters and co-cluster with TCRs [29], with unknown functional consequences. Previously, we and others have shown localized and synchronized TCR activation within the clusters at the plasma membrane of fixed and live T cells [30,31]. Such activation patterns are a hallmark of cooperative processes, and may occur through one of multiple different mechanisms [30] that may also be relevant for CARs in clusters. Moreover, TCR clustering is known to promote its activation [8]. Since CARs have not been designed to act in multimeric assemblies (esp. our CD138-CAR-Ts used here), the occurrence of CAR nanoclustering (e.g., Figure 2, Figure 3, Figure 4 and Figure 5) requires explanation; and any related effect on signaling remains to be described. Such an effect might occur, for instance, through trapping of kinases (ZAP-70 or Lck; as for TCRs), avidity effects upon binding of clustered ligands and more.

The segregation of TCRs from CD45 has been shown to occur in early T cell contacts with antigen-presenting cells (APCs), and to promote TCR-dependent cell activation [10,11]. Jung et al. [26,27] found that limited segregation of TCRs from CD45 occurs also on the surface of resting cells, as these molecules differentially localize at and around the tips of microvilli [26,27]. This limited segregation may explain the basal activation-independent segregation of CARs from CD45 upon engagement of non-target surfaces or cells in the images of live conjugates, as CARs and CD45 appear in adjacent clusters (e.g., Figure 2, Figure 3 and Figure 4). Thus, promoting a higher extent of ligand-dependent CAR clustering, or a more efficient segregation of CARs from CD45 (e.g., by a dedicated CAR design or choice of CAR target) could promote CAR-T efficacy. Thus, our results point to possible directions for enhancing CAR efficacy [32]. Moreover, changes in nanoscale clustering upon engagement with on-target cells precede markers of CAR-T activation and target cell killing. Hence, we propose that such nanoscale changes can facilitate CAR-T activation and serve as markers for predicting CAR-T efficacy, both at the single CAR-T cell level and at the CAR-T cell population level. Our imaging approach may also facilitate better understanding of molecular mechanisms that separate T cell from CAR-T signaling and activation [12,13].

## 5. Conclusions

Our results shed new light on the nanoscale organization of CARs on the surfaces of CAR-T cells. Specifically, we find that nanoscale CAR clustering and segregation from glycoproteins correlates with CAR-T cell activity and key effector functions. Thus, differences in CAR clustering and its segregation extent from CD45 at the interface between CAR-T cells and either on- or off-target cells could serve as mechanisms for target-dependent CAR-T activation, and sensitive markers for potential CAR-T efficacy and safety.

## Figures and Tables

**Figure 1 cells-12-02261-f001:**
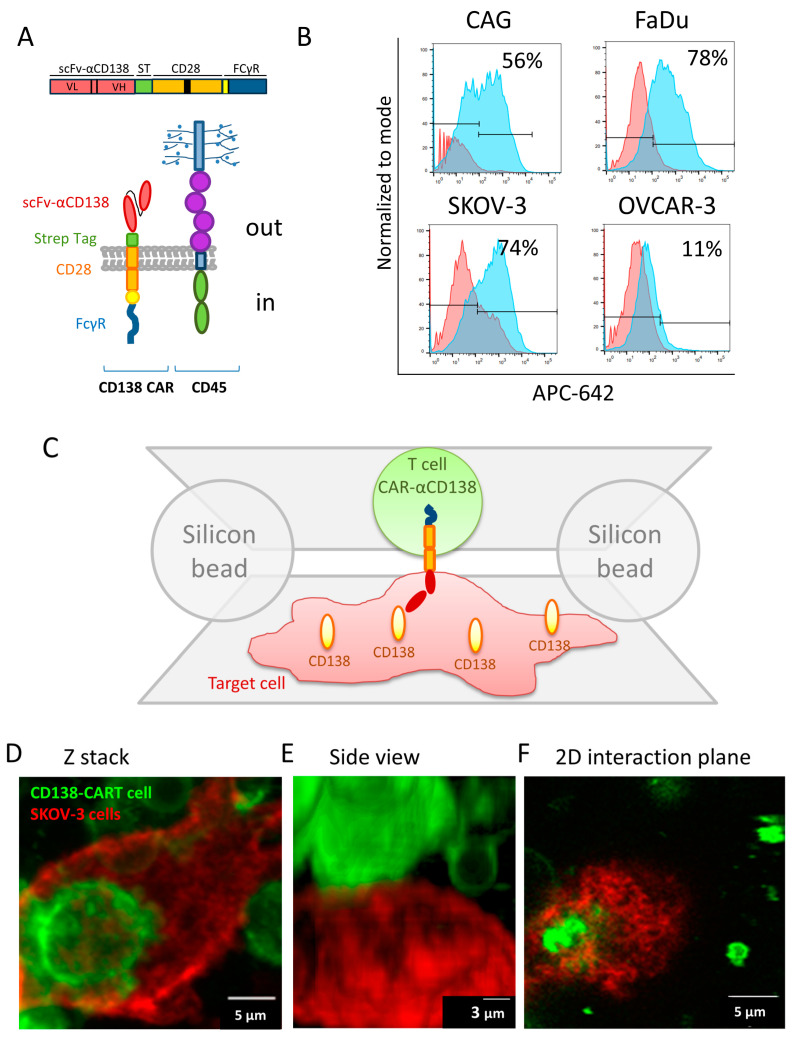
CAR design and cell-on-cell interaction method. (**A**) A schematic description of the CD138-CAR design and CD45. (**B**) FACS analysis for CD138 expression in different cell lines (CAG, FaDu, SKOV-3 and OVCAR-3). (**C**) A schematic representation of the cell-on-cell imaging approach. (**D**–**F**) Confocal imaging of two cells labelled with membrane dyes: CD138-CAR-T cell with DiO (green) on a SKOV-3 adherent cell, labelled with DiD (red). (**D**) Z-stack image of cell on cell. (**E**) Side view of the 3D confocal image in panel (**D**) of the two cells. (**F**) 2D plane of interaction between the two cells in panels (**D**,**E**). Scale-bars in panels (**D**,**F**) 5 µm or in panel (**F**) 3 µm.

**Figure 2 cells-12-02261-f002:**
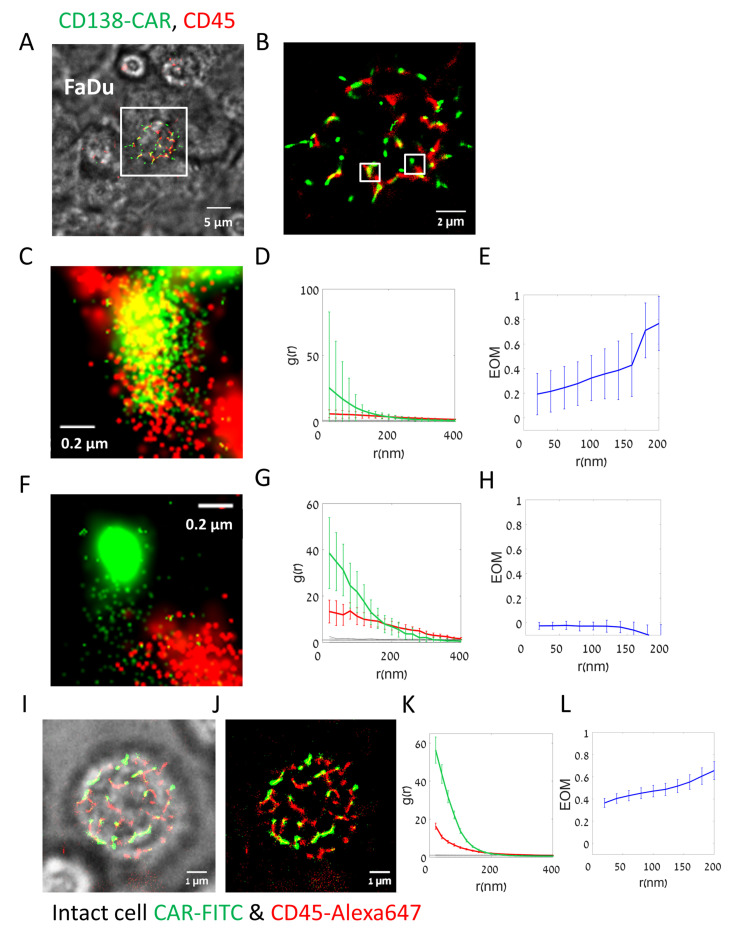
Two-color super resolution imaging on CD138-CAR-T upon interaction with target cell line. (**A**–**H**) Fluorescently labelled CD138-CAR-T cell on FaDu target cell. (**A**,**B**) Direct STORM (dSTORM) imaging of CD138-CAR-T cell on FaDu target cells. CD138 CAR-T was labelled with αSTREP tag-FITC (green) and αCD45-Alexa647 (red). Shown is an overlay of bright-field imaging merged with fluorescence two-color dSTORM imaging. (**B**) Zoom in on the fluorescent image of the CAR-T cell in panel (**A**). (**D**,**G**) Zoom in on the fluorescent image in two small areas (~4 µm^2^) inside the CAR-T cell in panel (**B**). (**E**,**F**) Pair-correlation function of CD138-CAR receptor (green) and CD45 molecules (red) within zoom regions in panel (**C**) (N = 4 regions), as represented in panels (**D**,**G**). (**G**,**H**) The extent of mixing (EOM) between CD138-CAR and CD45 within the zoom regions in panel (**C**) (N = 4 regions), as represented in panels (**D**,**G**). Scale-bars in panels A5 µm, in panel (**B**) 2 µm and in panels (**D**,**G**)—0.2 µm. Error bars are SEM. (**I**–**L**). Fluorescently labelled CD138-CAR-T intact cell, without a target cell. (**I**) Direct STORM (dSTORM) imaging of CD138-CAR-T intact cell, without a target cell on the bottom coverslip. CD138-CAR-T was labelled with αSTREP tag-FITC (green) and αCD45-Alexa647 (red). Bright-field imaging was merged with fluorescent two-color dSTORM imaging. (**J**) Fluorescent image only of the CAR-T cell in panel (**I**). (**K**) Pair-correlation function of CD138-CAR receptor (green) and CD45 molecules (red). (**L**) The extent of mixing (EOM) between CD138-CAR and CD45.

**Figure 3 cells-12-02261-f003:**
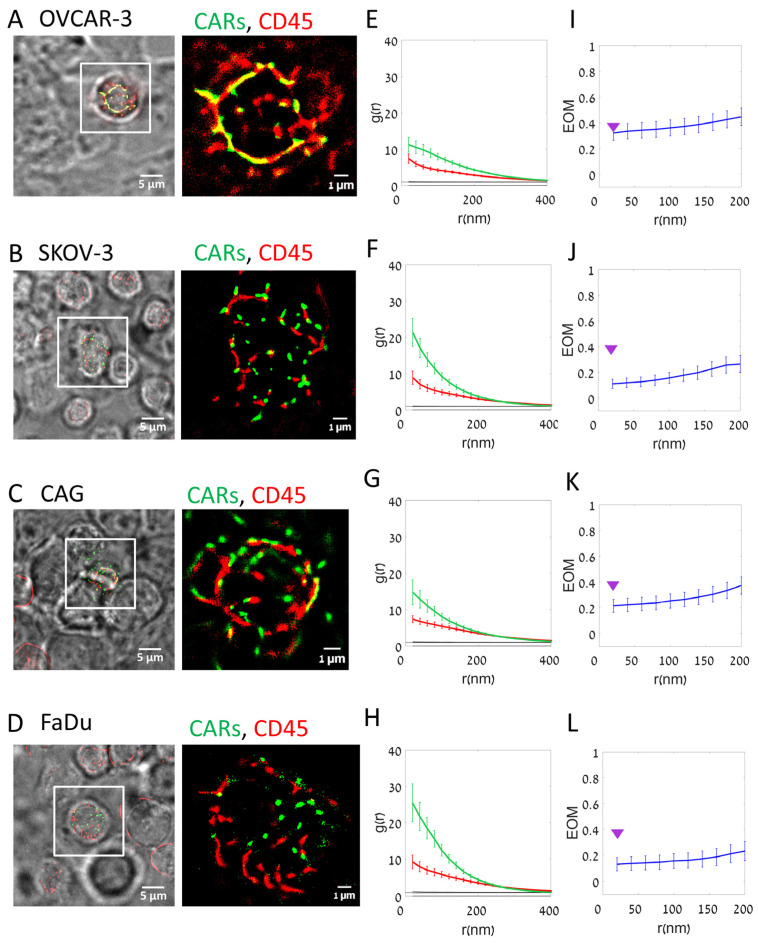
Segregation of CAR and CD45 on target cell lines (Fixed cells). (**A**–**D**) Fixed two-color dSTORM imaging of CD138-CAR-T cell on different cell lines: (**A**) OVCAR-3; (**B**) SKOV-3; (**C**) CAG; (**D**) FaDu. Visualizing CAR receptor (FITC-labeled, green) and CD45 molecules (Alexa647-labeled, red). Images from left to right: BF and fluorescence (left); fluorescence zoom image (right). (**E**–**H**) The average pair-correlation function of CD138-CAR molecules (green line) and CD45 molecules (red line) in T cell upon interaction with different cell lines: (**E**) OVCAR-3 (N = 36 cells); (**F**) SKOV-3 (N = 36); (**G**) CAG (N = 38); (**H**) FaDu (N = 27). (**I**–**L**) The extent of mixing (EOM) between CD138-CAR and CD45 upon interaction with different cell lines: (**I**) OVCAR-3 (N = 36); (**J**) SKOV-3 (N = 36); (**K**) CAG (N = 38); (**L**) FaDu (N = 27). For reference, purple arrowheads indicate EOM (20 nm) values of OVCAR-3. Scale-bars in panels (**A**–**D**)—5 µm. Error bars are SEM.

**Figure 4 cells-12-02261-f004:**
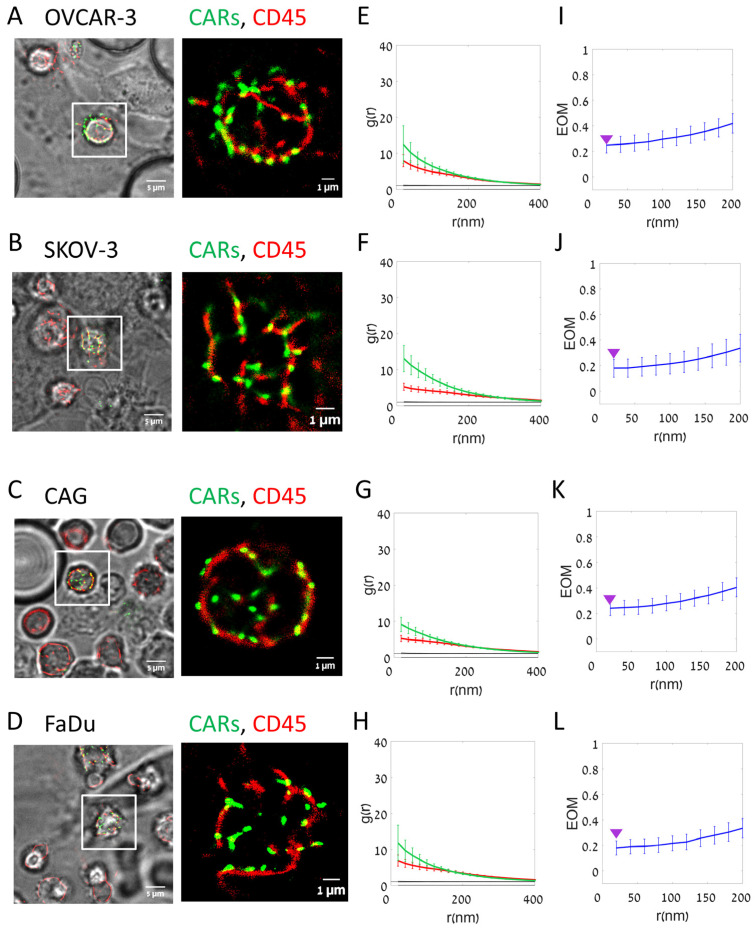
Segregation of CAR and CD45 on target cell lines (Live cells). (**A**–**D**) Live two-color dSTORM imaging of CD138-CAR-T cell on different cell lines: (**A**) OVCAR-3; (**B**) SKOV-3; (**C**) CAG; (**D**) FaDu. Visualizing CAR receptor (FITC-labeled, green) and CD45 molecules (Alexa647-labeled, red). Images from left to right: BF and fluorescence (left); fluorescence zoom image (right). (**E**–**H**) The average pair-correlation function of CD138-CAR molecules (green line) and CD45 molecules (red line) in T cell upon interaction with different cell lines: (**E**) OVCAR-3 (N = 22); (**F**) SKOV-3 (N = 13); (**G**) CAG (N = 25); (**H**) FaDu (N = 21). (**I**–**L**) The extent of mixing (EOM) between CD138-CAR and CD45 upon interaction with different cell lines: (**I**) OVCAR-3 (N = 22); (**J**) SKOV-3 (N = 13); (**K**) CAG (N = 25); (**L**) FaDu (N = 21). For reference, purple arrowheads indicate EOM (20 nm) values of OVCAR-3. Scale-bars in panels (**A**–**D**)—5 µm. Error bars are SEM.

**Figure 5 cells-12-02261-f005:**
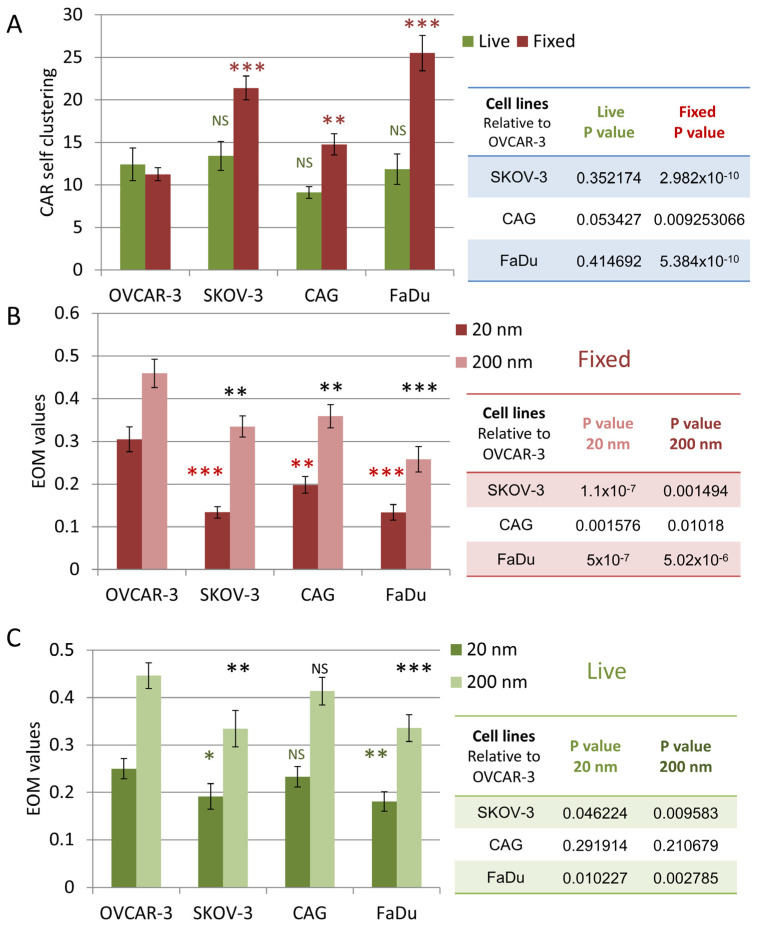
Extent of mixing and CAR self-clustering comparing in target vs. non-target cell lines. (**A**) A comparison of CD138-CAR self-clustering upon cell-on-cell interaction. Table on right summarizes *p*-values between target and non-target cells. (**B**,**C**) A comparison of EOM values on different cell lines at 20 nm and 200 nm in images of (**A**) fixed cells and (**B**) live cells. Error bars are SEM. Tables on right summarize *p*-values between target and non-target cells. Stars indicate *p*-values * *p* < 0.05, ** *p* < 0.01, *** *p* < 0.001. NS—non-significant.

**Figure 6 cells-12-02261-f006:**
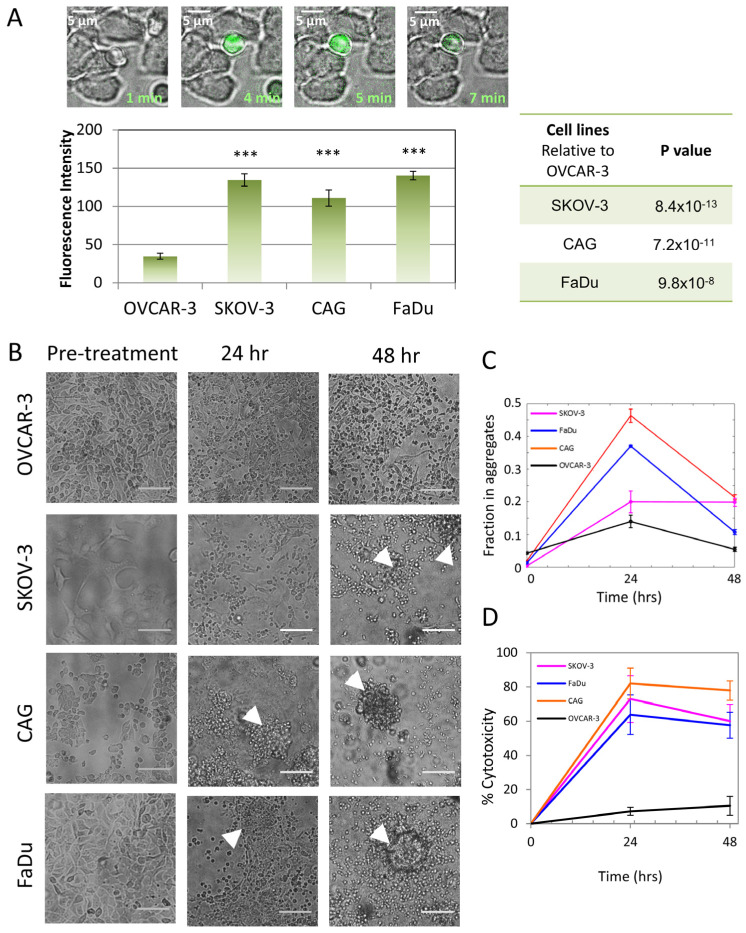
Early and late effector functions of CAR-T cell interaction with target cell lines. (**A**) Ca^++^ influx in CAR-T cells meeting different cell lines. Upper panel: Frames from movie of CD138-CAR-T round cells meeting CAG target adherent cell. In green Ca^++^ indicator. Lower panel: Mean fluorescence intensity 4 min after introduction CAR-T cells to target cell lines. Tables at right summarize *p*-values between target and non-target cells. (**B**) Bright-field images of cell lines’ viability after CD138-CAR-T cell introduction for 24 h and 48 h. White arrowheads point to representative cell aggregates. (**C**) Analysis of the fraction of cells in aggregates for all cells and conditions in panel (**B**). Cell aggregation indicates CAR-T-mediated killing of the specified cell lines. (**D**) MTT cytotoxicity assay of CAR-Ts engaging target/non-target cell lines. Scale-bars in panel (**A**)—5 µm, in panel (**B**)—10 µm. Error bars are SEM. Three stars indicate *p*-values < 0.001.

**Figure 7 cells-12-02261-f007:**
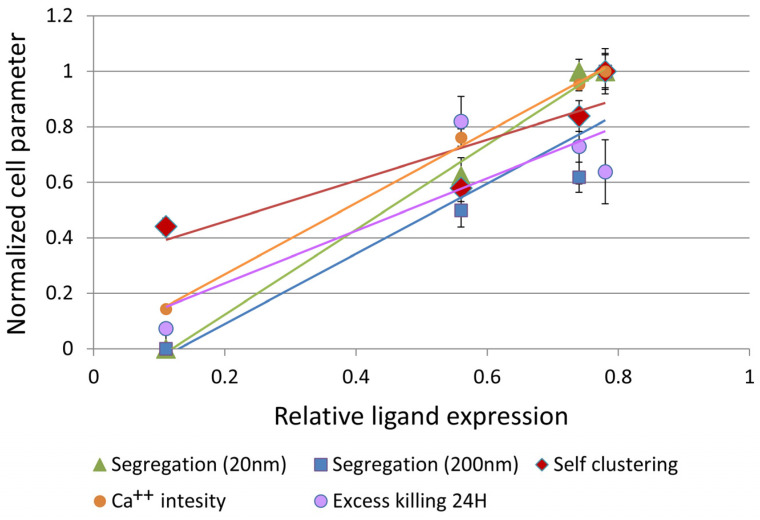
Correlation between normalized cell parameters and relative ligand expression. A plot of normalized cell parameters as a function of relative ligand expression (percent positive cells, measured by FACS). The parameters include: Segregation (i.e., 1-EOM) at 20 nm (calculated from Figure 5B); Segregation at 200 nm (calculated from Figure 5B); Self-clustering at 20 nm (Figure 5A); Ca^++^ influx (Figure 6A); Excess killing (i.e., killing extent of cell line—killing of OVCAR-3, calculated from the MTT assay in Figure 6D). Error bars are SEM.

**Table 1 cells-12-02261-t001:** Summary of Pearson correlation between normalized cell parameters.

Segregation (20 nm) *	0.996				
Segregation (200 nm) *	0.942	0.926			
Self-clustering	0.898	0.911	0.944		
Ca^++^ influx	0.999	0.991	0.935	0.876	
Excess killing 24 H (MTT)	0.862	0.832	0.724	0.557	0.887
	Ligand	Segregation (20 nm)	Segregation (200 nm)	Self-clustering	Ca^++^ influx

*—Segregation was defined as 1-EOM.

## Data Availability

The authors declare that the data supporting the findings of this study are available within the article and its Appendix A, or are available from: https://zenodo.org/deposit/8168494 accessed on 31 July 2023.

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
