# Peer review of "Nanoscale CAR Organization at the Immune Synapse Correlates with CAR-T Effector Functions"

_cells, 2023, doi:10.3390/cells12182261_

Round 1
Reviewer 1 Report (Previous Reviewer 1)
The authors thoroughly addressed all the concerns raised by the reviewer. The paper is interesting, sound, and can be published as it is. An error is on page one - maybe a font issue in one of the last lines.
Author Response
We thank the reviewers and editors once again for the consideration of our manuscript, and for their invaluable feedback.
Reviewer 2 Report (New Reviewer)
In Sajman et al, the authors discuss using super-resolution microscopy to interrogate the synapse formation between CAR and CD45 to explain efficacy against different tumors of various CD138 density. I believe that this technology adds considerable novelty to the CAR T cell field and thoroughly enjoyed reading it but needs some editing before I would consider it publication ready. I have broken down my comments to major and minor comments.
Major:
- The synapse difference between CAR and TCR has been a talking point for the CAR T cell field and there has been observation that the CAR itself makes a distinctly different synapse than TCR. See paper included: https://www.pnas.org/doi/10.1073/pnas.1716266115. I would be nice for you to discuss this in the discussion and suggest how your technology might add to the field for this.
- Secondly you discuss the optimal linker but only interrogate a single linker. I think to make this comment you should have tested at least a second linker length.
- While this might be beyond the scope of your paper, it might be interesting to look at the how different scFvs (high vs low affinity) impact synapse formation.
- Explaining a little more of how the CAR is stained would be helpful for the reader. I could have missed it but I was unsure as to how the streptavidin was binding to the CAR.
- For the primary cell and cell lines bit you should talk about how you cultured the cells. what media you used, FBS %, and how you got and made the CARs is needed.
Minor
- Keep the CAR T cell consistent throughout the paper. It is Car in the title when it should be CAR. You have CAR T cells, CAR-T cells and CAR-T in the paper. I would just pick one and use that.
- I would fix the first sentence in the abstract, “CARs are the forefront of clinical treatment of cancers using adoptive transfer of CAR‑T cells” says the same thing but nothing all at once.
- Check for formatting issues. Swirls showed up in the text and I didn’t know what they stood for.
- Adding a figure solely in the discussion is a bit unorthodox but I think it was a nice change in feel.
The quality was fine just needed to be read through a little more thouroughly as to avoid some confusion.
Author Response
We thank the reviewers and editors once again for the consideration of our manuscript. Below is our point-by-point response (Please note that changes in the text are detailed below but were incorporated without highlighting in the resubmitted text).
In Sajman et al, the authors discuss using super-resolution microscopy to interrogate the synapse formation between CAR and CD45 to explain efficacy against different tumors of various CD138 density. I believe that this technology adds considerable novelty to the CAR T cell field and thoroughly enjoyed reading it but needs some editing before I would consider it publication ready. I have broken down my comments to major and minor comments.
We appreciate the interest of the reviewer in our technology and study.
Major:
- The synapse difference between CAR and TCR has been a talking point for the CAR T cell field and there has been observation that the CAR itself makes a distinctly different synapse than TCR. See paper included: https://www.pnas.org/doi/10.1073/pnas.1716266115. I would be nice for you to discuss this in the discussion and suggest how your technology might add to the field for this.
We thank the reviewer for pointing out the reference, which we now cite. We suggest that our technology could help in further differentiating (T and CAR-T) cell level responses with high resolution and assist in optimizing CAR designs. We now mention in the Discussion that: "Our imaging approach may also facilitate better understanding of molecular mechanisms that separate T cell from CAR-T signaling and activation 12,13."
We further conclude that: "differences in CAR clustering and its segregation extent from CD45 at the interface between CAR-T cells and either on- or off-target cells could serve as mechanisms for target-dependent CAR-T activation, and sensitive markers for potential CAR-T efficacy and safety. "
- Secondly you discuss the optimal linker but only interrogate a single linker. I think to make this comment you should have tested at least a second linker length.
Following the reviewers' feedback, we have now taken out the comment on the optimal linker length and leave it to a future study.
- While this might be beyond the scope of your paper, it might be interesting to look at the how different scFvs (high vs low affinity) impact synapse formation.
We appreciate this suggestion by the reviewer and agree that this is certainly a parameter that would be of prime interest in a follow up study.
- Explaining a little more of how the CAR is stained would be helpful for the reader. I could have missed it but I was unsure as to how the streptavidin was binding to the CAR.
We apologize for not clarifying this point enough. The staining of the CAR was using a Strep tag included in its stem. We targeted this tag using an anti-Strep antibody labeled with the fluorophore FITC. We write in the results: “For dSTORM visualization, individual CARs carried a STREP tag, and were labelled using αSTREP-FITC antibody.” This is now better illustrated in the updated Fig. 1A.
- For the primary cell and cell lines bit you should talk about how you cultured the cells. what media you used, FBS %, and how you got and made the CARs is needed.
We thank the reviewer for this comment, and now provide the details and reference to the cell preparation and growing in the Methods.
“Cell lines (FaDu, SKOV-3, CAG and OVCAR-3) grow in RPMI + 10%FBS + 1% Pen/Strep. Primary UT and CART cells grow in RPMI + 10%FBS + 1%Pen/Strep + 100 units IL-2. CART cells were prepared according to a previous study [ https://pubmed.ncbi.nlm.nih.gov/33008840/]”
Minor
- Keep the CAR T cell consistent throughout the paper. It is Car in the title when it should be CAR. You have CAR T cells, CAR-T cells and CAR-T in the paper. I would just pick one and use that.
We have now corrected this issue throughout the text.
- I would fix the first sentence in the abstract, “CARs are the forefront of clinical treatment of cancers using adoptive transfer of CAR‑T cells” says the same thing but nothing all at once.
We have now modified this sentence, and simply state that “T cells expressing chimeric antigen receptors (CARs) are at the forefront of clinical treatment of cancers.”
- Check for formatting issues. Swirls showed up in the text and I didn’t know what they stood for.
We have now corrected this issue.
- Adding a figure solely in the discussion is a bit unorthodox but I think it was a nice change in feel.
We appreciate this comment and want to clarify that Fig. 7 shows a unifying analysis of the already shown results.
This manuscript is a resubmission of an earlier submission. The following is a list of the peer review reports and author responses from that submission.
Round 1
Reviewer 1 Report
Sajman et al. describe differences in CD45 and CAR organization within the immunological synapse between antigen presenting cells (APCs) and CAR-T cells. They use an interesting approach to bring both cell types in close contact and then conduct single molecule localization microscopy (SMLM) using dSTORM imaging. While the research on CAR-T cells in conjugation with APCs is of high interest and relevance - especially on the single molecule level – it is far from being trivial and care needs to be taken when interpreting results.
I have strong doubts on the interpretation of the results (not on the results per se!) based on the following points:
11. The authors cite publications about nanoclustering of immune-signaling relevant transmembrane and membrane-anchored proteins like the receptor tyrosine kinase Lck, the T-cell receptor or others (Refs. 3-5) and set their findings in terms of nanoclustering in context to these studies, but completely ignore more recent literature reporting the presence of overcounting artifacts 1-6. For the aforementioned proteins, the observed nanoclustering could be traced back to artifacts associated with the SMLM modality used. I highly recommend to interpret all “nanoclusters” in view of these new studies.
22. Related to 1, I can’t see any controls conducted to exclude overcounting artifacts. Mentioned ref. 31 does not describe strategies to avoid overcounting (e.g. a dSTORM experiment without any APCs, just an adherent coverslip).
33. It is not clear to me, which plane (i.e. interaction surface between CAR-T and APC) the authors chose for imaging. E.g. in figure 2 the target cell can be hardly observed in areas where the CAR and CD45 are observed (compare Figure 2a/b with c). Can the structures be elongated also in z? If this is the case (and the author later argue that this can happen), then the image would be a projection into the x-y plane and everything would have a clustered appearance.
44. What indeed was reported for studies between (CAR) T-cells and synthetic APCs (functionalized lipid bilayers) is the appearance of functional microclusters (studies by Dustin and many others). Looking at the timescale (“10 minutes”) and the temperature (“37°C”) I assume that all initial signaling already happened and the authors might look at these structures.
55. It is known for non-ratiometric Ca2+ indicators, that the signal strongly depends on cell shape (volume/area changes), plane of imaging, etc. . By looking at the y-axis of Figure 6a I wonder whether the slight increase in Ca2+ (from 220 to 300-340 units) is real of can also be interpretated as a different interaction between APC and CAR-T. Either the use of ratiometric dyes or other indicators further downstream would exclude this uncertainty.
1 Annibale, P., Vanni, S., Scarselli, M., Rothlisberger, U. & Radenovic, A. Identification of clustering artifacts in photoactivated localization microscopy. Nat Methods 8, 527-528 (2011). https://doi.org:nmeth.1627 [pii]10.1038/nmeth.1627
2 Annibale, P., Vanni, S., Scarselli, M., Rothlisberger, U. & Radenovic, A. Quantitative photo activated localization microscopy: unraveling the effects of photoblinking. PLoS One 6, e22678 (2011). https://doi.org:10.1371/journal.pone.0022678 PONE-D-11-04436 [pii]
3 Baumgart, F. et al. Varying label density allows artifact-free analysis of membrane-protein nanoclusters. Nat Methods (2016). https://doi.org:10.1038/nmeth.3897
4 Spahn, C., Herrmannsdorfer, F., Kuner, T. & Heilemann, M. Temporal accumulation analysis provides simplified artifact-free analysis of membrane-protein nanoclusters. Nat Methods 13, 963-964 (2016). https://doi.org:10.1038/nmeth.4065
5 Rossboth, B. et al. TCRs are randomly distributed on the plasma membrane of resting antigen-experienced T cells. Nat Immunol 19, 821-827 (2018). https://doi.org:10.1038/s41590-018-0162-7
6 Platzer, R. et al. Unscrambling fluorophore blinking for comprehensive cluster detection via photoactivated localization microscopy. Nat Commun 11, 4993 (2020). https://doi.org:10.1038/s41467-020-18726-9
Reviewer 2 Report
Sajman et al. in their work with the title “Nanoscale CAR organization at the immune synapse correlates with CAR-T effector functions” focus on an excellent question about the nanoscopic organisation of CARs on CAR-T cells forming contact with the target cells. Recent studies indicate that nanoscale organisation of immunoreceptors is critical for the proper immune response. In this respect, it is very important to understand how CARs are organised at the surface of CAR-T cells at nanoscale and in 3D. Even better, if this could be studied in living cells or in vivo (probably sci-fi). The key question of this work is thus superbly important. However, the authors apply their newly developed sample preparation system and single molecule localisation microscopy (SMLM) at the forefront of this work, overlooking that simpler methods may provide great information to learn about these cells at first. Presented data lack quality, inappropriate methods are applied and conclusion are overinterpreted (see below for more details). Thus, I cannot recommend this work for publication.
The system used by Sajman et al. involved immobilisation of target cells on the coverslip and of CAR-T cell on another slide which is placed on top of target cells. The distance of the two surfaces is kept constant at 20 um using beads. This is an interesting approach to increase formation of immunological synapses which are notoriously inefficient when T cells are applied in suspension. However, presented data (apart from Fig. 6A) show cells in contact but these do not necessarily have to involve formation of the immunological synapse (IS). No evidence is provided that the IS was formed. Thus, the presented results do not represent CAR organisation in the synapse. Moreover, no data using intact CAR-T cells are presented. The study thus lacks comparison of CAR organisation before and after formation of the IS.
Clusters formed by CARs, as shown in Figures 2-4, are of > 0.5 um in diameter. Microclusters of this size should be ‘visible‘ using confocal or wide-field microscopies. Unfortunately, no data using these direct imaging approaches are provided.
In this work, CD45 is shown in clusters. Previous studies (refs. 11, 27 in the ms and Franke et al. Comm Biol 2022) coherently demonstrate that CD45 is excluded from the tips of microvilli, but otherwise exhibits a rather random distribution on T-cell surface. This indicates that current setup somehow interferes with normal distribution of surface molecules.
Overall quality of SMLM images is very low. It would be interesting to compare RAW data with the presented images. Current directions in professional scientific publishing do not accept ‘data available upon reasonable request’ statements anymore. Please, provide all raw data used for this publication (e.g. via ZENODO or B2SHARE). Related is also the use of FITC-labelled streptavidin for SMLM. This fluorophore is inappropriate for this method. I agree that some laboratories managed to successfully employ unusual fluorophores for SMLM. Even more then, seeing raw data would be very helpful to understand if this is the case.
ZOOMed SMLM images, which should indicate nanoscopic organisation of CARs (and CD45?) – Fig. 2D and G - are from the edge of cells. Using 2D imaging, this area is difficult to interpret and is usually removed from postprocessing analysis.
On page 8 authors mention that CAR clusters are dominantly formed by dimers or trimers. With localisation precision of 40 nm, it is impossible to distinguish molecular organisation of clusters. Moreover, authors used outdated cluster analysis method – pair correlation. Current development offers much better tools to analyse receptor clusters using SMLM – see Nieves et al. Nat Meth 2023, https://doi.org/10.1038/s41592-022-01750-6 ).
The interface between CAR-T cells and target cells is highly three-dimensional. The newly developed system by authors is more efficient, but does not change anything on this fact. Thus, 3D imaging should be used instead of 2D for CAR nanoscopic organisation analysis.
Methods used for data analysis are NOT properly described in the Methods sections. All details, including potential macros for ImageJ should be provided.
Viability assay based on cell aggregation is unacceptable in 2023. Please, use one of the standard approaches (e.g., fluorescence microscopy using probes such as 7AAD combined with Cell Green or variants of MTT assay).
Reviewer 3 Report
Please see the attachment.
